# The Role of Gender in Patients with Borderline Personality Disorder: Differences Related to Hopelessness, Alexithymia, Coping Strategies, and Sensory Profile

**DOI:** 10.3390/medicina59050950

**Published:** 2023-05-15

**Authors:** Andrea Amerio, Antimo Natale, Giovanni Battista Gnecco, Alessio Lechiara, Edoardo Verrina, Davide Bianchi, Laura Fusar-Poli, Alessandra Costanza, Gianluca Serafini, Mario Amore, Andrea Aguglia

**Affiliations:** 1Department of Neuroscience, Rehabilitation, Ophthalmology, Genetics, Maternal and Child Health (DINOGMI), Section of Psychiatry, University of Genoa, 16132 Genoa, Italy; andrea.amerio@unige.it (A.A.); giovannibattistagnecco@gmail.com (G.B.G.); lechiara.alessio@gmail.com (A.L.); edoardo.verrina@hotmail.com (E.V.); gianluca.serafini@unige.it (G.S.); mario.amore@unige.it (M.A.); 2IRCCS Ospedale Policlinico San Martino, 16132 Genoa, Italy; 3Department of Psychiatry, Adult Psychiatry Service (APS), University Hospitals of Geneva (HUG), 1205 Geneva, Switzerland; antimo.natale@yahoo.it (A.N.); alessandra.costanza@unige.ch (A.C.); 4Department of Mental Health and Pathological Addictions, Lavagna Local Health Authority, 16033 Lavagna, Italy; davidebianchi.md@gmail.com; 5Department of Brain and Behavioral Sciences, University of Pavia, 27100 Pavia, Italy; laura.fusarpoli@unipv.it; 6Department of Psychiatry, Faculty of Medicine, Geneva University (UNIGE), 1211 Geneva, Switzerland; 7Department of Psychiatry, Faculty of Biomedical Sciences, University of Italian Switzerland (USI), 6900 Lugano, Switzerland

**Keywords:** borderline personality disorder, gender differences, substance use, coping, alexithymia, sensory profile, hopelessness

## Abstract

*Background and Objectives*: Gender differences are poorly investigated in patients with borderline personality disorder (BPD), although they could be useful in determining the most appropriate pharmacological and non-pharmacological treatment. The aim of the present study was to compare sociodemographic and clinical characteristics and the emotional and behavioral dimensions (such as coping, alexithymia, and sensory profile) between males and females with BPD. *Material and Methods*: Two hundred seven participants were recruited. Sociodemographic and clinical variables were collected through a self-administered questionnaire. The Adolescent/Adult Sensory Profile (AASP), Beck Hopelessness Scale (BHS), Coping Orientation to Problems Experienced (COPE), and Toronto Alexithymia Scale (TAS-20) were administered. *Results*: Male patients with BPD showed more involuntary hospitalizations and greater use of alcohol and illicit substances compared to females. Conversely, females with BPD reported more frequent medication abuse than males. Furthermore, females had high levels of alexithymia and hopelessness. Regarding coping strategies, females with BPD reported higher levels of “restraint coping” and “use of instrumental social support” at COPE. Finally, females with BPD had higher scores in the Sensory Sensitivity and Sensation Avoiding categories at the AASP. *Conclusions*: Our study highlights gender differences in substance use, emotion expression, future vision, sensory perception, and coping strategies in patients with BPD. Further gender studies may clarify these differences and guide the development of specific and differential treatments in males and females with BPD.

## 1. Introduction

Borderline personality disorder (BPD) is a severe mental condition characterized by a pervasive and enduring pattern of instability in relationships, behaviors, emotions, and self-image [1]. This condition is associated with a high risk of suicidal behaviors, illicit substance use, and psychiatric comorbidities, with a significant burden on patients, family members, and the healthcare system [2,3]. Indeed, patients with BPD are frequently admitted to acute psychiatric wards [4] and no univocal pharmacological treatment and psychosocial interventions are provided as the first-line treatment. Medications should be limited to critical situations and polypharmacy is not recommended [4,5].

BPD is the most common personality disorder; epidemiological studies have estimated a prevalence that ranges between 1 and 3.9% in the general population [6]. In clinical populations, BPD prevalence increases up to 10% in psychiatric outpatients and between 15% and 25% in inpatients [7,8]. Despite the Diagnostic and Statistical Manual of Mental Disorders, Fifth Edition, Text Revision (DSM 5-TR) [9] indicating that BPD is diagnosed mainly in females (about 75%), only a few studies have investigated gender clinical differences in this population. In general, females affected by BPD show more suicidal behaviors, self-harm, affective instability, and feelings of emptiness, compared to males [10,11,12]. Indeed, the clinical presentation in females is characterized by a higher prevalence of diagnostic symptoms and increased symptom severity, with more internalizing clinical presentations, with higher rates of anxiety, depression, and symptoms of post-traumatic stress disorder (PTSD). Conversely, males showed more externalizing symptoms, with a higher prevalence of illicit substance use and impulsive behaviors [13]. Finally, other authors found that females are at a higher risk of engaging in promiscuous sexual relationships [14]. This evidence could support the hypothesis about the existence of gender-related clinical differences in patients with BPD.

As previously reported, suicidality is one of the main features of this disorder. Indeed, patients with BPD attempt suicide, and 10% of cases result in completed suicides [15]. As is known, suicidality strictly relates to hopelessness, defined as a cognitive pattern based on negative expectations about the future [16], and patients with BPD often showed higher feelings of hopelessness compared to other psychiatric populations [17]. Few studies have explored this dimension in patients with BPD, underlining that meaning in life could play a role as a protective factor against suicidal behaviors [18,19,20,21]. Other studies have also reported that individuals with BPD showed affective instability and altered coping strategies [22]. In particular, it has been shown that patients with BPD have trouble recognizing and identifying their own feelings and distinguishing between emotions and somatic sensations, a condition known as alexithymia [23].

Alexithymia is an externally oriented cognitive style that allows people to avoid recognition and awareness of emotion [24]. A recent study found higher levels of alexithymia in patients with BPD compared to healthy subjects; this finding points out the importance of uncovering this affective trait in BPD. The same authors reported an association between alexithymic traits and deficits in perceptions of facial emotions, leading to more severe emotional intensity and tension due to misperceptions in social situations [25], the most destabilizing factor identified in patients affected by BPD [26]. A recent study has underlined that alexithymia may partially mediate the relationship between childhood adversities and affective lability and impulsivity; moreover, affective lability and impulsivity may mediate the association between alexithymia and a diagnosis of BPD, thus attributing to alexithymia a significant role in patients with BPD [27].

Regarding coping strategies, it was found that adults with BPD exhibited a lower set of coping strategies than the general population, and higher coping inflexibility [28]; more specifically, females with BPD showed the tendency to present emotion-oriented coping strategies, while males showed more avoidance-oriented coping strategies [29], rather than task-oriented coping styles [30]. This finding was confirmed in a recent study, conducted by Carlson and coworkers, in which emotion-oriented coping and social diversion-oriented coping mediated the association between BPD symptoms and social dysfunction in female patients; conversely, coping did not mediate the association between BPD symptoms and social role dysfunction in males [31].

Another important aspect concerning the characterization of patients with BPD is the alteration of emotional and sensory processes, which has been seen to play a primary role in the pathophysiology of various psychiatric disorders including BPD, especially in association with maladaptive coping techniques [32]. Furthermore, individuals with BPD are polarized to a subset of both sensory-sensitive and sensory-avoiding [33], having a clinical correlation with negative mood (i.e., depressed, angry, and anxious) [34] and alexithymic traits [35].

The clinical features of BPD appear very complex, especially regarding coping strategies, emotional and cognitive processes, and sensory profiles. However, very few studies have highlighted gender differences in patients with BPD. The present study was designed to reach a better comprehension of these core features of BPD, with the specific purpose of identifying gender-related differences. We expected to find gender differences in several clinical dimensions investigated, such as coping strategies, alexithymia or sensory profile that could help clinicians to implement targeted and personalized non-pharmacological intervention, particularly useful for this population. The main hypothesis is that female patients with BPD could report higher levels of alexithymia, hopelessness, and, consequently, different patterns of pathological use of medications, while males could have more externalizing symptoms with a higher prevalence of alcohol, illicit substances, and impulsive behaviors.

## 2. Materials and Methods

### 2.1. Study Design and Participants

The study is a cross-sectional investigation including two hundred seven participants who met the Diagnostic and Statistical Manual of Mental Disorders, Fifth Edition (DSM-5) criteria for BPD. Our sample was recruited at the Section of Psychiatry, Department of Neuroscience, Rehabilitation, Ophthalmology, Genetics, Maternal and Child Health (DiNOGMI), IRCCS Ospedale Policlinico San Martino, University of Genoa (Italy), from 1 April 2019 to 30 September 2022.

Inclusion criteria were the following: age ≥ 18 years, ability to understand and willing to sign informed consent, and a primary diagnosis of BPD. Exclusion criteria consisted of severe neurological disorders (e.g., epilepsy, cognitive impairment, or genetic syndromes), presence of cognitive deficits causing linguistic and comprehension problems, pregnant patients, and re-hospitalizations in case of multiple admission of the same patient. In the present study, we excluded 46 patients who presented with neurological and cognitive disorders or had missing data (Figure 1).

During hospitalization, clinical evaluations were carried out by expert clinicians and carefully reviewed by a senior psychiatrist (with at least 10 years of clinical experience in an inpatient clinical setting and with suicidal behaviors). Potential participants were provided with an in-depth explanation of the study objectives and procedures and an opportunity to ask questions. Written informed consent was obtained prior to their recruitment. The study was designed in agreement with the guidelines from the Declaration of Helsinki [36] and was approved by the local Ethical Review Board.

### 2.2. Assessment and Procedures

Recruiters conducted a clinical interview with each patient, collecting several anamnestic data, such as gender, current age, educational level, nationality, family history for psychiatric disorders, suicidal ideation, number and prevalence of suicide attempts, number of hospitalizations, number of involuntary admissions, substance use and medication abuse (current and lifetime).

All participants enrolled completed a battery of assessments: Adolescent/Adult Sensory Profile (AASP), Beck Hopelessness Scale (BHS), Coping Orientation to Problems Experienced (COPE), and Toronto Alexithymia Scale (TAS-20).

The AASP [37] is a 60-item questionnaire developed to estimate how people respond to different sensory stimuli. This test includes 6 sections: taste and smell processing, movement processing, visual processing, touch processing, activity level, and auditory processing. Items are divided equally into the four fundamental traits: Low Registration, Sensation Seeking, Sensory Sensitivity, and Sensation Avoiding.

The BHS is a 20-item dichotomous (true-false) scale designed to assess current negative expectations about the future, evaluating three different dimensions: feelings about the future, loss of motivation, and expectations [38]. It is accepted as a measure of higher risk for suicidal behaviors for psychiatric patients and general population [39].

The COPE [40] is a self-administered psychometric instrument consisting of 60 items in which patients indicate, on a scale of one to four, how many times a specific coping strategy is used when experiencing significant stress. The results indicate three dimensions as follows: problem-focused coping, emotion-focused coping, and potentially maladaptive strategies.

The TAS-20 [41] is a 20-item self-report scale measure of alexithymia. The total score can vary from 20 to 100. This scale yields a total score and subscale score for identifying feelings, describing feelings, and externally oriented thinking. The cutoff scores for the TAS-20 are as follows: ≤51, no alexithymia; 52–60, possible alexithymia; ≥61, alexithymia.

### 2.3. Statistical Analysis

Continuous and categorical variables were presented as means and standard deviations (SD) or frequency and percentage, respectively. The normal distribution was evaluated using the Kolmogorov–Smirnov test, before applying statistical analyses. First, the sample was divided according to gender (males and females). Chi-squared tests were performed to evaluate differences between categorical variables, while Student’s *t*-test for independent samples was performed to evaluate differences between continuous variables. Receiver operating characteristic (ROC) curve analysis was used to measure the diagnostic value of our significant differences to bivariate analyses (we reported only the main findings). Finally, a logistic regression analysis was performed in order to investigate possible sociodemographic and clinical variables associated with gender. Statistical analyses were performed using the Statistical Package for the Social Sciences (SPSS) for Windows 25.0 (IBM Corp., Armonk, NY, USA). The results were considered statistically significant for *p* < 0.05.

## 3. Results

### 3.1. Gender Differences in Sociodemographic and Clinical Characteristics

Regarding sociodemographic variables, 140 patients were females (67.6%) and 67 were males (32.4%), with a mean age of 33.87 ± 13.57 years. The majority of participants were single (68.6%). All sociodemographic and clinical data of patients enrolled in the study, with a specific comparison between genders, are shown in Table 1.

Statistical analysis revealed some gender-related differences. Males reported a higher prevalence of lifetime involuntary hospitalizations than females (41.8% vs. 27.9%, *p* = 0.045). As for illicit substances, current consumption of alcohol (52.2% vs. 34.3%; *p* = 0.014), both lifetime (70.1% vs. 54.3%; *p* = 0.004) and current substance use (52.2% vs. 31.4%; *p* = 0.004) were more frequent in males than females. Conversely, medication abuse was more common in females than in males with BPD, both lifetime (40% vs. 25.4%; *p* = 0.039) and currently (27.9% vs. 13.4%; *p* = 0.021).

### 3.2. Gender Differences in Sensory Profile, Alexithymia, Suicidal Attitudes, and Coping Strategies

We compared the total scores reported at the AASP, TAS-20, BHS, and COPE to evaluate gender differences in sensory profile, alexithymia, suicidal attitudes, and coping strategies, respectively. Regarding sensory profile characteristics, females reported higher scores in both the Sensory Sensitivity (*p* = 0.025) and Sensory Avoiding (*p* = 0.035) subscales of the AASP. Additionally, TAS scores were higher in females than males (*p* = 0.005), with 57.1% of females, obtaining a total score above the cut-off of 61, compared to 40.3% of males (*p* = 0.023). In the analysis of the BHS, 70% of females had a score above 9, compared to 53.7% of males (*p* = 0.023). All results are shown in Table 2.

As for coping strategies, measured using the COPE, females reported using two problem-focused coping strategies more often than males: restraint and use of instrumental social support (*p* = 0.022 and *p* = 0.026, respectively). Results on COPE gender difference are integrally displayed in Table 3.

### 3.3. ROC Curve

ROC curve was performed to assess the diagnostic value of significant findings at bivariate analyses, shown in Figure 2. The area under the ROC curve of AASP_sensory sensitivity and AASP_sensation avoiding was 0.590 and 0.594, respectively. Furthermore, the area under the ROC curve of COPE_restraint coping and COPE_use of instrumental social support was 0.619 and 0.598, respectively. Lastly, the area under the ROC curve of hopelessness was 0.620, while alexithymia was 0.621.

### 3.4. Logistic Regression Analysis

Table 4 reports the results of the backward logistic regression analysis to evaluate factors independently associated with gender in patients with BPD. Characteristics that remained significantly associated with the male gender in BPD were involuntary admissions and current alcohol use disorder. Conversely, the female gender remained significantly associated with current medication abuse, higher levels of hopelessness, as indicated by BHS ≥ 9, higher levels of alexithymia, as indicated by TAS-20 ≥ 61; and higher restraint and use of instrumental social support coping strategies.

## 4. Discussion

The present study aimed to identify gender differences in patients with BPD. Specifically, we explored sociodemographic and clinical characteristics, as well as several emotional and behavioral dimensions, such as coping strategies, alexithymia, sensory profile, and hopelessness.

Our results showed that males with BPD were more likely to use alcohol and illicit substances; conversely, medication abuse was more typical of females with BPD. Indeed, the literature has reported that males with BPD exhibit more frequently externalizing behaviors (including substance use disorder, and antisocial and narcissistic traits) while females present more internalizing behaviors (such as depression and anxiety) [12,14,42]. However, published studies have mainly focused on alcohol and illicit substance use, while little is known about medication abuse. Our findings revealed that medication abuse was significantly higher in females; this result was further validated by the regression analysis that indicated current medication abuse as a factor significantly associated with the female gender. This finding indicates that research in this field should be implemented in order to avoid the prescription of drugs that can lead to abuse (such as benzodiazepines) and to limit the total number of drugs prescribed to female patients with BPD. Additionally, our sample revealed a higher number of involuntary hospitalizations in males than in females with BPD due to a greater presence of externalizing behaviors. In fact, the increased antisocial behaviors, including illicit substance use, with the consequent tendency toward aggressiveness, could justify the need to resort to involuntary hospitalization in the acute phases of the disease.

In our sample, females reported higher levels of alexithymia compared to males with BPD. Several studies have indicated that individuals with BPD show difficulties in accurately describing their emotional reactions and have higher levels of alexithymia than healthy controls. Furthermore, alexithymia and depression scores have been reported to predict BPD status [25,43,44]. In particular, difficulty in identifying and describing emotions rather than externally oriented thinking seems to be present in BPD [45]. Of note is that patients with BPD associated with the presence of alexithymia experience a worsening in BPD-related symptoms, such as behavioral impulsivity, suicidality, and interpersonal dysfunction [46,47]. Additionally, interpersonal trauma, conflictual relationships with caregivers, childhood adversities, and impairments in emotion regulation represent shared etiopathological factors between alexithymia and BPD [48,49,50]. Evidence has also suggested that alexithymia might be a mediator between trauma and emotion dysregulation in the context of BPD [27,51,52]. It is worth considering that alexithymia appears to be associated with several psychiatric and somatic conditions and, therefore, it is not a unique feature of BPD; for example, two meta-analyses have shown that alexithymia is also associated with PTSD [53] and eating disorders [54]. However, to our knowledge, no studies have considered gender differences in the context of alexithymia and BPD. The presence of higher levels of alexithymia in females could be explained by the transdiagnostic nature of alexithymia. Indeed, it is known that female patients present more frequently with comorbid PTSD and eating disorders than males with BPD [14]. Therefore, alexithymia could act as a common substrate for these entities. Further studies are needed to corroborate this hypothesis and evaluate the possible relationship between alexithymia and specific comorbid patterns in females with psychiatric disorders.

In our sample, female patients were more likely to report BHS scores equal to or higher than 9 as compared to male patients with BPD. Of note is that a BHS score above 9 has been used in previous studies to indicate an increased risk of depression and suicidal behaviors in inpatient setting [55,56,57]. This result is in line with the literature indicating that females with BPD have an increased risk of depression and suicidal behaviors [12,42]. Additionally, elevated levels of hopelessness were found to be more frequently associated with the female gender, a history of childhood trauma, and high levels of alexithymia [57]. Thus, our finding may help to clarify the tendency of females with BPD to experience internalizing depressive symptoms; providing insights and implementing the monitoring of self-harm and suicidal behaviors are needed [58,59].

The findings on coping strategies and sensory profiles also point in the same direction. In fact, our results showed that females with BPD reported higher scores in two subscales of COPE related to “problem-focused coping”, specifically in the categories “restraint coping” and “use of instrumental social support”. In the description of the COPE scale, Carver defined restraint coping as the tendency of “waiting until an appropriate opportunity to act presents itself, holding oneself back, and not acting prematurely”; the use of instrumental support refers instead to “seeking advice, assistance, or information” [40]. In essence, females with BPD would be less prone to externalize the act with the tendency to ask for help. These findings further confirm that females with BPD are more prone to ask for help and tend to internalize through forms of anxiety and depression. It is interesting to note that no significant differences were found in the other two subscales of COPE, i.e., “emotional-focused coping” and “potentially disadaptive strategies”, indicating that these two traits represent core features of BPD, regardless of gender. Regarding sensory profile, our data show that females with BPD had higher levels in the “sensory sensitivity” and “sensation avoiding” domains. Specifically, “sensory sensitivity” predisposes individuals to be more sensitive to sensory inputs and reflects a greater functioning of the behavioral inhibition system [60]. Individuals with sensory sensitivity report increased emotional, biological, and stress responsiveness to sensory stimuli [60,61]. This implies that highly sensitive people are likely to be overwhelmed by sensory stimuli and to experience the world as highly unpredictable and anxiety-provoking. Sensation avoiding occurs when individuals respond to sensory stimuli with a behavioral response characterized by withdrawal from the stimulus and active avoidance of it, accompanied by feelings of discomfort and anxiety [62]. Therefore, our results showed that females with BPD have a profile characterized by greater sensitivity to sensory stimuli, which are abnormally experienced and may cause overwhelm, anxiety, and subsequent avoidance behaviors. This finding reflects the daily clinical practice and it is frequently found in the experiences described during structured interviews by patients with BPD.

Despite the interesting findings, our study presents several limitations. First, since our study has a cross-sectional design, it was not possible to draw any inference on the temporal or causal relationship between the variables considered. Second, several clinical variables (such as comorbidities with anxiety and mood disorders and a history of childhood trauma) that could influence the clinical dimensions studied were not included in the analyses due to the large number of missing values. Third, data collection took place in a single research center and inpatient unit, thus including patients with acute symptomatology.

## 5. Conclusions

Our findings revealed that females may present a distinct profile in terms of several clinical dimensions. In fact, female individuals with BPD have a marked hypersensitivity to external stimuli, which overwhelm them and generate anxiety and avoidance. They present levels of alexithymia with a greater tendency towards hopelessness, which is reflected in an increased risk of depression and suicidal behaviors [21], particularly if neurologic language impairments are present [63]. Indeed, females with BPD are more prone to medication abuse and tend not to act too prematurely and seek help from social support facilities. These data reflect the tendency of females with BPD to engage in internalizing behaviors. On the contrary, males with BPD were more prone to substance use and had higher rates of involuntary hospitalizations, confirming the trend toward externalizing behaviors.

Our data highlight gender differences in the context of BPD, suggesting the need to adopt differential diagnostic and therapeutic strategies aimed at correctly implementing non-pharmacological approaches in patients with BPD. In fact, a differentiation according to gender may help patients to avoid relapses of the disease and favor intervention on specific harmful and dangerous behaviors for patients affected by BPD, and also reduce the socioeconomic impact of care.

## Figures and Tables

**Figure 1 medicina-59-00950-f001:**
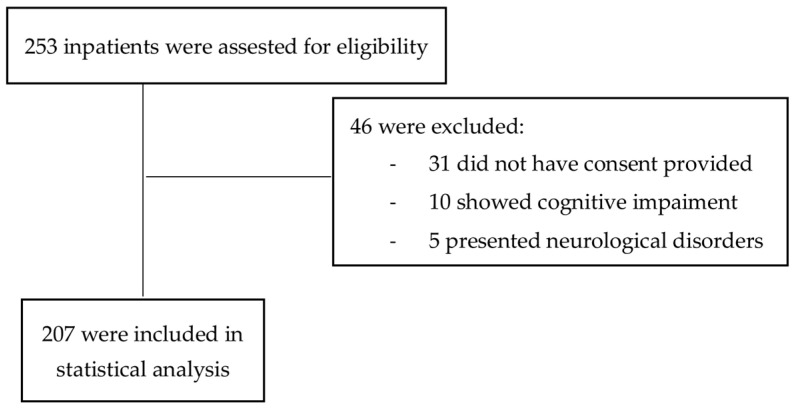
Flow diagram (application of the eligibility criteria of the study).

**Figure 2 medicina-59-00950-f002:**
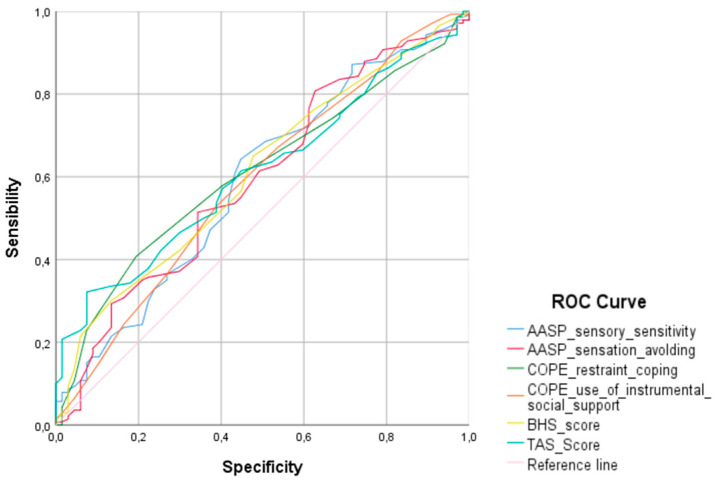
ROC curve analysis performed to assess the diagnostic value of mean platelet volume, neutrophil-lymphocyte ratio and platelet-lymphocyte ratio.

**Table 1 medicina-59-00950-t001:** Sociodemographic and clinical characteristics of males and females with BPD.

*n* (%) or Mean ± SD	Total Sample (*n* = 207)	Males (*n* = 67)	Females (*n* = 140)	X^2^/t	*p*
Current age	33.87 ± 13.57	34.72 ± 13.41	33.46 ± 13.68	0.620	0.536
Marital status				4.171	0.244
Single	142 (68.6)	49 (73.1)	93 (66.4)		
Married	35 (16.9)	13 (19.4)	22 (15.7)		
Divorced	29 (14.0)	5 (7.5)	24 (17.1)		
Widowed	1 (0.5)	0 (0.0)	1 (0.7)		
Educational level	11.57 ± 3.36	11.28 ± 3.15	11.71 ± 3.45	−0.849	0.397
Nationality				1.758	0.185
Italian	182 (87.9)	56 (83.6)	126 (90.0)		
Others	25 (12.1)	11 (16.4)	14 (10.0)		
Number of hospitalizations	3.29 ± 3.32	3.15 ± 3.21	3.35 ± 3.38	−0.406	0.685
Involuntary admissions	67 (32.4)	28 (41.8)	39 (27.9)	4.019	0.045
Suicide ideation	112 (54.1)	33 (49.3)	79 (56.4)	0.939	0.332
Suicide attempts	130 (59.7)	40 (59.7)	90 (64.3)	0.408	0.523
Number of suicide attempts	2.41 ± 2.54	2.55 ± 3.29	2.34 ± 2.14	0.425	0.672
Alcohol use lifetime	120 (58.0)	44 (65.7)	76 (54.3)	2.411	0.120
Alcohol use current	83 (40.1)	35 (52.2)	48 (34.3)	6.081	0.014
Substance use lifetime	123 (59.4)	47 (70.1)	76 (54.3)	4.729	0.030
Substance use current	79 (38.2)	35 (52.2)	44 (31.4)	8.316	0.004
Medication abuse lifetime	73 (35.3)	17 (25.4)	56 (40.0)	4.247	0.039
Medication abuse current	48 (23.2)	9 (13.4)	39 (27.9)	5.293	0.021

**Table 2 medicina-59-00950-t002:** Gender differences in sensory profile, alexithymia, and attitudes toward suicide in participants with BPD.

*n* (%) or Mean ± SD	Males (*n* = 67)	Females (*n* = 140)	*t*-Test	*p*
AASP Low Registration	26.91 ± 11.46	28.70 ± 10.96	−1.083	0.280
AASP Sensation Seeking	33.63 ± 11.37	35.24 ± 10.12	−1.028	0.305
AASP Sensory Sensitivity	32.18 ± 11.80	37.21 ± 16.31	−2.259	0.025
AASP Sensation Avoiding	31.73 ± 11.09	35.24 ± 11.08	−2.128	0.035
Toronto Alexithymia Scale-20	58.25 ± 10.30	63.44 ± 13.03	−2.856	0.005
Presence of Alexithymia (TAS-20 ≥ 61)	27 (40.3)	80 (57.1)	5.149	0.023
Beck Hopelessness Scale	8.82 ± 3.99	10.39 ± 4.09	−2.597	0.010
Presence of Hopelessness (BHS ≥ 9)	36 (53.7)	98 (70.0)	5.254	0.022

AASP: Adolescent/Adult Sensory Profile; BHS: Beck Hopelessness Scale; SD: standard deviation; TAS-20: Toronto Alexithymia Scale.

**Table 3 medicina-59-00950-t003:** Gender differences in coping strategies in participants with BPD.

Mean ± SD	Males (*n* = 67)	Females (*n* = 140)	*t*-Test	*p*
*COPE Problem-focused coping*				
Active coping	10.57 ± 2.73	10.53 ± 2.49	0.101	0.920
Planning	9.96 ± 2.79	10.37 ± 2.63	−1.045	0.297
Suppression of competing activities	9.96 ± 2.59	10.24 ± 2.49	−0.768	0.443
Restraint coping	9.13 ± 1.87	9.85 ± 2.19	−2.305	0.022
Use of instrumental social support	9.67 ± 2.89	10.66 ± 3.00	−2.236	0.026
*COPE Emotion-focused coping*				
Use of social-emotional support	9.76 ± 2.80	9.74 ± 2.76	0.045	0.964
Positive reinterpretation and growth	10.42 ± 4.76	9.84 ± 3.26	1.017	0.310
Acceptance	10.06 ± 2.67	10.10 ± 2.72	−0.100	0.920
Humor	8.67 ± 2.71	8.91 ± 3.12	−0.546	0.586
Venting of emotions	9.99 ± 3.02	10.72 ± 2.69	−1.770	0.078
Turning to religion	8.51 ± 3.23	8.39 ± 3.64	0.233	0.816
*COPE Potentially disadaptive strategies*				
Denial	9.18 ± 2.65	9.26 ± 3.16	−0.175	0.861
Behavioral disengagement	9.39 ± 2.65	9.23 ± 2.35	0.438	0.662
Alcohol and drug disengagement	9.04 ± 3.30	9.24 ± 3.19	−0.414	0.680
Mental disengagement	9.78 ± 2.55	10.27 ± 2.35	−1.381	0.169

COPE: Coping Orientation to Problems Experienced.

**Table 4 medicina-59-00950-t004:** Stepwise logistic backward regression to evaluate factors independently associated with gender in patients with BPD.

Variables	*p*	OR	95% CI
Involuntary admissions lifetime	0.038	0.484	0.244–0.962
Current alcohol use	0.006	0.402	0.208–0.775
Current medication abuse	0.015	3.056	1.244–7.505
Presence of hopelessness (BHS ≥ 9)	0.010	2.582	1.253–5.321
Presence of alexithymia (TAS-20 ≥ 61)	0.039	2.053	1.037–4.065
COPE restraint coping	0.049	1.174	1.001–1.377
COPE use of instrumental social support	0.005	1.201	1.058–1.363

BHS: Beck Hopelessness Scale; COPE: Coping Orientation to Problems Experienced; CI: confidence interval; OR: odds ratio; TAS-20: Toronto Alexithymia Scale.

## Data Availability

Data that support the findings of this study and materials are available from the corresponding author, upon request.

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
