# Peer review of "The Role of Gender in Patients with Borderline Personality Disorder: Differences Related to Hopelessness, Alexithymia, Coping Strategies, and Sensory Profile"

_medicina, 2023, doi:10.3390/medicina59050950_

Round 1

Reviewer 1 Report

This is a good conducted study. Many points are being discussed, which show the rigor of the authors. I think that this experimental study can open perspectives for clinicians. The article presents a research conducted to investigate the gender differences in terms of some psychological factors including coping strategies, alexithymia, sensory profile and hopelessness in patients with BPD. There are, however, some points that might contribute to the improvement of the manuscript. They are listed below:
Introduction:

-          Please rewrite the research literature regarding variables in correct order based on your title, for example, you have discussed Hopelessness before coping strategies in patients with BPD. This will confuse the readers. 

-          Please elaborate more on the research literature regarding your variables including coping strategies, Alexithymia, Sensory Profile and Hopelessness in male and females with BPD by using more recent research works, to highlight the novelty and necessity of your work.

-          clarify why you have selected mentioned variables to compare between male and females with BPD?(this would indicate the importance of your work)

-          Add the complete research aims or the proposed hypothesis of the project at the end of introduction.

Materials and method:

-          Please mention the type of your study in methodology.

-          Add a subtitle for measures section.

-          What was the inclusion and exclusion criterion of selecting subjects?

Author Response

Reviewer 1

This is a good conducted study. Many points are being discussed, which show the rigor of the authors. I think that this experimental study can open perspectives for clinicians. The article presents a research conducted to investigate the gender differences in terms of some psychological factors including coping strategies, alexithymia, sensory profile and hopelessness in patients with BPD. There are, however, some points that might contribute to the improvement of the manuscript.

Introduction:

Question 1: Please rewrite the research literature regarding variables in correct order based on your title, for example, you have discussed Hopelessness before coping strategies in patients with BPD. This will confuse the readers. 

  • Reply to Question 1: Many thanks for this comment. We apologize for the confusion. The title has been changed according to order presented in the Introduction.

Question 2: Please elaborate more on the research literature regarding your variables including coping strategies, Alexithymia, Sensory Profile and Hopelessness in male and females with BPD by using more recent research works, to highlight the novelty and necessity of your work.

  • Reply to Question 2: Many thanks for this comment. Several recent studies have been added in the Introduction.

Question 3: clarify why you have selected mentioned variables to compare between male and females with BPD? (this would indicate the importance of your work)

  • Reply to Question 3: Many thanks for this comment. A statement has been added in the Introduction to clarify the rationale of investigating the variables of interest

Question 4: Add the complete research aims or the proposed hypothesis of the project at the end of introduction.

  • Reply to Question 4: Many thanks for this suggestion. A statement has been added at the end of the introduction with the proposed hypotheses.

Materials and method:

Question 5: Please mention the type of your study in methodology.

  • Reply to Question 5: Thank you for this suggestion. According to the Reviewer’ suggestion, the type of study design has been added.

Question 6: Add a subtitle for measures section.

  • Reply to Question 6: Thank you for this indication. A subtitle of each section has been added.

Question 7: What was the inclusion and exclusion criterion of selecting subjects?

  • Reply to Question 7: Many thanks for this comment. Inclusion and exclusion criteria have been included in the 2.1 section (study design and participants).

Reviewer 2 Report

I read the article titled with " The role of gender in patients with borderline personality disorder: differences related to coping strategies, alexithymia, sensory profile, and hopelessness". My suggestions are below. Please clarify the main contributions/novelties of this study for readers to better understand this paper.

1. The contributions are not stated clearly in the Introduction section and abstract.

2. Please highlight some limitations of the proposed method and some error cases of the model.

3. The schematic of the proposed Method should be given and explained.

4. Some abbreviations do not have explanations. The abbreviation table should be attached to the article. (as an appendix)

5. Recent studies (2020 ,2021,2022,2023) should be added to the references.

6. I suggest including ROC curves to assess the discriminatory role between healthy and sick patients.

7. I recommend performing other tests of statistical inference, such as linear regression models (beta coefficients) and logistic regressions (OR) to strengthen the observed comparative findings.

8. Inclusion and exclusion criteria for the study, should be described after reporting the study design.

9. The English and presentation can be improved.

Author Response

Reviewer 2

I read the article titled with " The role of gender in patients with borderline personality disorder: differences related to coping strategies, alexithymia, sensory profile, and hopelessness". My suggestions are below. Please clarify the main contributions/novelties of this study for readers to better understand this paper.

Question 1: The contributions are not stated clearly in the Introduction section and abstract.

  • Reply to Question 1: Many thanks for this comment. The abstract has been stated clearly and the Introduction section has been reorganized, also in line with comments raised by Reviewer 1.

 Question 2: Please highlight some limitations of the proposed method and some error cases of the model.

  • Reply to Question 2: Thank you for this comment. We have now added a sentence in the Discussion regarding the limitation related to the cross-sectional design.

Question 3: The schematic of the proposed Method should be given and explained.

  • Reply to Question 3: Many thanks for this indication. A flow diagram has been presented.

Question 4: Some abbreviations do not have explanations. The abbreviation table should be attached to the article. (as an appendix)

  • Reply to Question 4: Many thanks for this indication. An appendix with all abbreviation has been provided.

Question 5: Recent studies (2020, 2021, 2022, 2023) should be added to the references.

  • Reply to Question 5: Many thanks for this comment. Several recent studies have been added.

Question 6: I suggest including ROC curves to assess the discriminatory role between healthy and sick patients.

  • Reply to Question 6: Many thanks for this comment. The ROC curve was performed and provided.

Question 7: I recommend performing other tests of statistical inference, such as linear regression models (beta coefficients) and logistic regressions (OR) to strengthen the observed comparative findings.

  • Reply to Question 7: Thank you very much for this comment. In Paragraph 3.4. we have reported a backward logistic regression analysis to evaluate factors independently associated with gender in patients with BPD to strengthen the findings of our study.

Question 8: Inclusion and exclusion criteria for the study, should be described after reporting the study design.

  • Reply to Question 8: Many thanks for this comment. Inclusion and exclusion criteria have been included in the 2.1 section (study design and participants).

Question 9: The English and presentation can be improved.

  • Reply to Question 9: Many thanks for this indication. The English language has been revised by a native speaker.

Round 2

Reviewer 2 Report

Most of the corrections suggested in the article did not. Also, corrections are not marked in the article.

1. The contributions are not stated clearly in the Introduction section and abstract.

2. Please highlight some limitations of the proposed method and some error cases of the model.

3. The schematic of the proposed Method should be given and explained.

4. Some abbreviations do not have explanations. The abbreviation table should be attached to the article. (as an appendix)

5. Recent studies (2020 ,2021,2022,2023) should be added to the references.

6. I suggest including ROC curves to assess the discriminatory role between healthy and sick patients.

7. I recommend performing other tests of statistical inference, such as linear regression models (beta coefficients) and logistic regressions (OR) to strengthen the observed comparative findings.

8. Inclusion and exclusion criteria for the study, should be described after reporting the study design.

9. The English and presentation can be improved.

Author Response

I read the article titled with " The role of gender in patients with borderline personality disorder: differences related to coping strategies, alexithymia, sensory profile, and hopelessness". My suggestions are below. Please clarify the main contributions/novelties of this study for readers to better understand this paper.

Question 1: The contributions are not stated clearly in the Introduction section and abstract.

  • Reply to Question 1: Many thanks for this comment. The abstract has been stated clearly and the Introduction section has been reorganized, also in line with comments raised by Reviewer 1.

Question 2: Please highlight some limitations of the proposed method and some error cases of the model.

  • Reply to Question 2: Thank you for this comment. We have now added a sentence in the Discussion regarding the limitation related to the cross-sectional design.

Question 3: The schematic of the proposed Method should be given and explained.

  • Reply to Question 3: Many thanks for this indication. A flow diagram has been presented.

Question 4: Some abbreviations do not have explanations. The abbreviation table should be attached to the article. (as an appendix)

  • Reply to Question 4: Many thanks for this indication. An appendix with all abbreviation has been provided.

Question 5: Recent studies (2020, 2021, 2022, 2023) should be added to the references.

  • Reply to Question 5: Many thanks for this comment. Several recent studies have been added.

Question 6: I suggest including ROC curves to assess the discriminatory role between healthy and sick patients.

  • Reply to Question 6: Many thanks for this comment. The ROC curve was performed and provided.

Question 7: I recommend performing other tests of statistical inference, such as linear regression models (beta coefficients) and logistic regressions (OR) to strengthen the observed comparative findings.

  • Reply to Question 7: Thank you very much for this comment. In Paragraph 3.4. we have reported a backward logistic regression analysis to evaluate factors independently associated with gender in patients with BPD to strengthen the findings of our study.

Question 8: Inclusion and exclusion criteria for the study, should be described after reporting the study design.

  • Reply to Question 8: Many thanks for this comment. Inclusion and exclusion criteria have been included in the 2.1 section (study design and participants).

Question 9: The English and presentation can be improved.

  • Reply to Question 9: Many thanks for this indication. The English language has been revised by a native speaker.